# An Optimized Microwave-Assisted Digestion Method to Analyze the Amino Acids Profile of *Quisqualis Fructus* from Different Planted Origins

**DOI:** 10.3390/foods13111645

**Published:** 2024-05-24

**Authors:** Lei Dai, Lin Yang, Yiwu Wang, Yan Li, Jianing Zhao, Shuxiang Pan, Yaxuan Li, Dan Yang, Dan He

**Affiliations:** 1Chongqing Research Center for Pharmaceutical Engineering, College of Pharmacy, Chongqing Medical University, Chongqing 400016, China; 2021120834@stu.cqmu.edu.cn (L.D.); 2023121867@stu.cqmu.edu.cn (J.Z.); 2023121968@stu.cqmu.edu.cn (S.P.); 18099910025@163.com (Y.L.); 18183103454@139.com (D.Y.); 2Chongqing Pharmaceutical Preparation Engineering Technology Research Center, Chongqing Medical and Pharmaceutical College, Chongqing 401331, China; yyanglin925@163.com (L.Y.); ly20031079@163.com (Y.L.); 3Experimental teaching center, Chongqing Medical University, Chongqing 400016, China; willwyw@cqmu.edu.cn

**Keywords:** *Quisqualis Fructus*, microwave-assisted digestion, response surface methodology, amino acids, chemometric, fingerprint

## Abstract

This study aims to establish a rapid and convenient microwave-assisted digestion method for sample pretreatment to determine amino acid profiles in natural products. This method was applied to analyze the amino acid profiles of *Quisqualis Fructus* (QF) from different planted origins. The microwave-assisted digestion conditions were optimized by a response surface methodology (RSM), and 17 amino acids in different planted origins of QF were determined by an automatic amino acid analyzer according to the optimized digestion conditions. The contents of 17 amino acids in QF from different planted origins were further analyzed by fingerprint and chemometric analysis. The temperature of microwave digestion at 167 °C, time of microwave digestion at 24 min, and a solid–liquid ratio of 46.5 g/mL was selected as the optimal digestion conditions. The total content of 17 amino acids in QF from different planted origins ranged from 71.88 to 91.03 mg/g. Amino acid composition and nutritional evaluation indicated that the content of medicinal amino acids was higher than aromatic amino acids. The results of fingerprint analysis reflected that the similarity between the 16 batches of QF ranged from 0.889 to 0.999, while chemometrics analysis indicated amino acid content in QF varied from different planted origins, and six important differential amino acids were screened. Compared with the traditional extraction method, microwave-assisted digestion with response surface optimized has the advantages of rapidity, convenience, and reliability, which could be used to study the amino acid profiles in natural products. The amino acid profile of QF indicated that it has a rich medicinal nutritional value. Different planted origins of QF have a high degree of similarity and could be effectively distinguished by chemometric analysis.

## 1. Introduction

*Quisqualis Fructus* (QF) is a deciduous vine-like shrub plant belonging to the Combretaceae family, which is widely distributed in China, India, Malaysia, and the Philippines [1,2] below. In China, the plant is mainly planted in the southwestern region, such as Chongqing, Sichuan, Yunnan, Guangxi, and other places. Among them, the production of QF in Hechuan, Chongqing, is the largest, and QF in Tongliang, Chongqing, is recognized as a national protected product with geographical indications [3,4,5]. Previous research revealed that it is rich in medicinal ingredients such as trigonelline, quisqualic acid, and ellagic acid, which are commonly used to kill worms, and eliminate stagnation, strengthen the spleen, and stop diarrhea in people [6,7]. It also contains a variety of nutritional ingredients such as fatty acids, polysaccharides, amino acids, and trace elements [8,9,10]. QF is a deep-rooted plant with relatively long aboveground vines, usually cultivated in the forest edge of valleys, streamsides, and sunny roadsides in plain areas at an altitude of 1200 m. It is suitable for sunny, humid, convenient drainage and irrigation, and deep soil layers of sandy soil cultivation [3].

The determination of amino acid content is one of the most effective indicators for the quality evaluation of natural products [11,12,13]. Amino acids can be categorized into essential and non-essential amino acids according to their functions. They have the role of promoting the growth and differentiation of cells in the body, maintaining normal physiological functions, and ensuring the energy metabolism of the body. Disorders of amino acid metabolism are closely related to many diseases, such as metabolic diseases, cardiovascular diseases, immune diseases, and cancer [14,15]. Currently, it has been reported that eight free amino acids, including proline, threonine, and others are found to be present in QF, using thin-layer chromatography (TLC). However, there is no comprehensive qualitative and quantitative research on the amino acid profile of QF [10]. Domestic standards for the determination of amino acid content in natural product samples often refer to the National Standard for Food Safety of the Determination of Amino Acids in Food (GB 5009.124-2016). Foreign official standards mainly refer to the International Organization for Standardization’s (ISO) 13904:2016, the American Association of Analytical Chemists’ (AOAC) 994.12 (1997), and the European Commission (EC) 152/2009 [16]. The above methods stipulate that the sample for determination of free amino acids in proteins should be hydrolyzed by hydrochloric acid (6 mol/L) at 110 °C for 20–24 h. In recent years, the microwave digestion method has been gradually used in the pretreatment for the determination of amino acids in proteins with the advantages of accelerating the rate of hydrolysis, simple processing, and reliable results. Importantly, there is no significant difference in the results of the determination of the content of each amino acid compared with the traditional digestion method [17,18].

The methods of amino acids quantitative determination mainly include pre-column derivatization–high performance liquid chromatography (HPLC), post-column derivatization–automatic amino acid analyzer, liquid chromatography–mass spectrometry (LC-MS), gas chromatography–mass spectrometry (GC-MS), and nuclear magnetic resonance (NMR) [19,20,21,22]. Among them, an automatic amino acid analyzer has the advantages of high efficiency, precision, and automation, which is one of the most widely used methods [23]. Response surface methodology (RSM) could be used to reduce the number of experiments and method costs by efficiently designing reaction conditions, which is widely used in industrial, agricultural, and medical fields [24]. When microwave-assisted digestion conditions of amino acid determination were optimized through RSM, optimization results could be rapidly evaluated by determination of the content of total amino acid. Fingerprint and chemometrics analysis is a mathematical and statistical method that is an important tool for site differentiation, processing, concocting, source identification, and growth stage identification of medicinal plants [25,26]. When analyzed by both methods, samples from different origins could be analyzed for quality consistency by fingerprint analysis and compared for origin variability by chemometric analysis.

In our work, an optimal microwave digestion method was established by response surface methodology (RSM) for the pre-digestion treatment of amino acid determination in QF. The method was subsequently compared with other traditional amino acid extraction methods. Seventeen amino acids in QF from different planted origins were determined and analyzed after methodological verification. Except for the evaluation of the nutritional value of QF, the content comparison of amino acids in QF from different planted origins was carried out by fingerprint and chemometrics analysis. The above research provided a fast digestion pretreatment idea for the determination of amino acids in natural products and further elaborates the amino acid profile and origin variability of traditional Chinese medicine of QF.

## 2. Materials and Methods

### 2.1. Chemicals and Reagents

The amino acids mixture standard solution (Type H, 2.5 μmol/mL) including aspartic acid (Asp), threonine (Thr), serine (Ser), glutamic acid (Glu), proline (Pro), glycine (Gly), alanine (Ala), cysteine (Cys), valine (Val), methionine (Met), isoleucine (Ile), leucine (Leu), tyrosine (Tyr), phenylalanine (Phe), lysine (Lys), histidine (His) and arginine (Arg), and the Ninhydrin Coloring Solution Kit were purchased from FUJIFILM Wako Pure Chemical Co. (Osaka, Japan). The MCI BUFFERTM PH-KIT buffer system was purchased from Mitsubishi Chemical Co. (Tokyo, Japan). Hydrochloric acid (AR) was obtained from Chongqing Chuandong Chemical Group Co. (Chongqing, China). Sodium citrate was obtained from Shanghai Macklin Biotechnology Co. (Shanghai, China). Ultrapure water was produced using a Milli-Q system (Millipore, Milford, MA, USA).

### 2.2. Sample Information

A total of 16 batches of QF were collected from the agricultural products trading market in each producing area, including Chongqing (CQ1-CQ4), Sichuan (SC1-SC4), Yunnan (YN1-YN4), and Guangxi (GX1-GX4). The fruits were dried and stored in a dry and cool place in the laboratory. It was identified as the dried ripe fruit of QF by Associate Professor Wang Jian, College of Traditional Chinese Medicine, Chongqing Medical University.

### 2.3. Extraction Method of Amino Acids

The amino acid extraction methods were observed, such as traditional acid hydrolysis, heating reflux, ultrasonic extraction, and microwave-assisted digestion extraction methods. The amino acid extraction method of traditional acid digestion was carried out by referring to National Food Safety Standard of the Amino Acid Determination in Food (GB 5009.124-2016) [27], and this approach was carried out by taking 0.2 g sample in an oven at 110 °C with 10 mL hydrochloric acid (6 mol/L) for 22 h; the amino acids extraction method of water bath reflux was performed refer to the determination of free amino acids in plants (GB/T 30987-2020) [28], and this approach was performed by taking 0.2 g sample in 10 mL boiling water for 10 min; The amino acids extraction method of ultrasonic extraction was performed by referring to the Chinese Pharmacopoeia (2020 Edition) [29], and this approach was made by taking 0.2 g sample in an ultrasonic environment with 10 mL 50% methanol for 30 min. The amino acids extraction method of microwave-assisted digestion was performed referring to the previous literature [30], and this approach was processed by taking a 0.2 g sample in a microwave environment with 10 mL hydrochloric acid (6 mol/L) for 20 min and were optimized as follows.

The optimized microwave-assisted digestion was performed on a WX-6000 microwave digestion instrument from Shanghai Yiyao Technology Development Co. (Shanghai, China). A total of 0.2 g powder of QF was accurately weighed and placed into a PTFE microwave digestion tube after passing through the 80 mesh sieve. The above tube was added with 9.3 mL hydrochloric acid solution (6 mol/L) and filled with argon gas for 2 min subsequently. The optimized microwave digestion conditions are presented in Table 1. The hydrolysate solution was rinsed with ultrapure water 3 times and diluted in a 50 mL volumetric flask after being cooled to room temperature. The diluted solution was filtered through a 0.22 μm filter membrane. A filter solution of about 0.5 mL was accurately measured into a 1.5 mL EP tube to evaporate, drying under a vacuum at 60 °C. The residue was dissolved in 1.0 mL sodium citrate buffer (pH = 2.2) with the c(Na^+^) of 0.2 mol/L. The test solution was obtained after vortexed well and stored in a refrigerator at 4 °C.

### 2.4. Instrument Conditions

The amino acids determination was performed on an LA8080 automatic amino acid analyzer from Hitachi High-Tech Science Co. (Tokyo, Japan). The analytical column was a Hitachi HPLC Packed column (Ion exchange resin #2622, 4.6 mm × 60 mm, 3 μm) and an ammonia filter column (Hitachi strongly acidic cation-exchange resin #2650 L, 4.6 mm × 40 mm) was used for removal of ammonia in eluent. The MCI BUFFERTM PH-KIT buffer system was transmitted at a flow rate of 0.40 mL/min, including solutions PH-1 (B1), PH-2 (B2), PH-3 (B3), PH-4 (B4), water (B5), PH-RG (B6), while column temperature was maintained at 57 °C and the injection volume was 10 μL. Ninhydrin(R1), ninhydrin buffer solution (R2), and 5% ethanol (R3) were delivered at a flow rate of 0.35 mL/min; after producing color at a reaction temperature of 135 °C, the resultant derivatives were measured by UV detection at two wavelengths of 570 and 440 nm simultaneously. The derivative of Pro was detected at 440 nm due to a low response at 570 nm.

### 2.5. Experimental Design of Box–Behnken Design (BBD)

The three-factor and three-level response surface of Box–Behnken design (BBD) in RSM was designed to optimize microwave digestion conditions. Previous literature has reported the appropriate microwave conditions that were used for milk powder digestion by the same manufacturer of the digestion apparatus [30]. Meanwhile, we justified the upper and lower limits of these conditions by referring to the relevant literature reports that were used for natural product digestion [16,31]. Finally, the level of influencing factors were determined, including microwave heating temperature (150, 165, and 180 °C), microwave heating time (10, 20, and 30 min), and solid–liquid ratio (25, 50, and 70 g/mL). The total amino acid content (mg/g) served as an indicator for investigation. The level of experimental design is shown in Table 2.

### 2.6. Software

Design Expert 13.0 software (State-East corporation, Minneapolis, MN, USA) was applied to optimize the microwave-assisted digestion condition of amino acid determination. The Box–Behnken design (BBD) in RSM was performed, involving three factors and three levels. Analysis of variance (ANOVA) was performed to compare the significance using Fisher’s least significant difference (LSD) between different influencing factors. Interaction and model diagnosis displayed the relationship between influencing factors and to evaluate the reliability of the model, respectively. Origin 2022 (Originlab corporation, Hampton, MA, USA) and IBM SPSS Statistics 20 software (International Business Machines Corporation, Armonk, NY, USA) were conducted to visualize measurement results and analyze the significance of different producing areas. The Chinese Medicine Fingerprint Similarity Evaluation Software (2012 version) (National Pharmacopoeia Committee, Beijing, China) and SIMCA 14.1 software (Umetrics, Umea, Sweden) were carried out for quality consistency evaluation and origin difference analysis.

## 3. Results and Discussion

### 3.1. Comparison of Amino Acid Extraction Methods

Compared with heating by heat conduction and convection of traditional acid digestion, microwave-assisted digestion makes polar molecules (water and protein) vibrate by the action of a high-frequency magnetic field, causing friction between molecules and heating. Therefore, it has the advantages of fast, convenient, safe, and uniform heating [17,32]. Our research has proved again that there was no significant difference in the total amino acid content between the microwave-assisted acid digestion method and the traditional acid digestion method. As shown in Figure 1a,b. Except for Met and Arg, the content of other amino acids determined by the ultrasonic extraction method was significantly lower than the acid digestion method. The total amino acid content of ultrasonic extraction accounts for about 1/5 of the acid digestion method, indicating the main amino acids in QF were bound amino acids. It also reflected that the content of free amino acids in QF was low, and Met and Arg were more easily degraded under strong acid conditions. As displayed in Figure 1c. The content of each amino acid by water bath reflux extraction was significantly lower than other methods. The total amount of amino acid content obtained by water bath reflux extraction was about 1/20 of the acid digestion method, as presented in Figure 1d. Through the above research, the microwave-assisted digestion method with the shorter digestion time and the same reliable results was selected for sample digestion pretreatment. Response surface optimization was subsequently used to refine the microwave-assisted digestion conditions to determine more scientific. The chromatograms of the mixed standard solution and the comparison of the amino acid contents of the four extraction methods are shown in Figure 1e,f.

### 3.2. Optimization of Microwave Digestion Conditions by BBD

#### 3.2.1. The Results of Experimental Design

Response surface optimization refers to the analysis of the regression relationship between test indicators and multiple test factors in the treatment of multi-factor quantitative tests. Owing to its convenience and reliability, it was widely used to optimize the extraction process of natural products [33]. The BBD experiments in RSM were carried out to optimize microwave digestion conditions at different levels, including microwave heating temperature (A), microwave heating time (B) and solid–liquid ratio (C). A total of 17 trial experiments were designed and conducted by BBD model. The BBD experiments and results are displayed in Table 3.

#### 3.2.2. Analysis of ANOVA

According to the results of the BBD model at three factors and three levels, 17 sets of experimental data were used in the configure analysis. In order to simplify the regression model and avoid overfitting, the quadratic polynomial regression model was used to express the relationship between the total amino acid content of QF and different variables, including microwave digestion temperature (A), microwave digestion time (B), and solid–liquid ratio (C). The code equation for the yield of total amino acid was Y = 84.21 + 1.61A + 1.86B − 4.25C − 0.1145AB + 4.69AC + 4.08BC − 4.29A^2^ − 1.49B^2^ − 7.28C^2^. The significant regression model (R2 = 0.9809, *p* < 0.0001) and insignificant lack of fit (*p* > 0.6027) of ANOVA analysis indicated that the unknown factors had little interference with the experimental results, and the model could be used for analysis and prediction. The results are summarized in Table 4. From the F value, it could be seen that the influence factors of the three factors on the comprehensive score from large to small were the ratio of solid to liquid (C) > microwave digestion time (B) > microwave digestion temperature (A). From the *p*-value, it could be seen that the influence of factors B, C, AC, BC, A2, and C2 was extremely significant (*p* < 0.01), the influence of factors A was significant (*p* < 0.05), and the influence of factors AB and B2 were no significant (*p* > 0.05). Satisfactory results with reliable predictive ability and high goodness of fit could be obtained from the common quadratic polynomial model; therefore, the other models were not adopted in our research.

#### 3.2.3. Diagnosis and Interaction Analysis

The externally studentized residuals of the regression model analysis of variance showed that the test results were close to the predicted results, indicating that the error of the test results was small (Figure 2A). The test results were distributed on both sides of the linear fitting equation of the predicted results, indicating that the fitted linear equation has high reliability (Figure 2B). The influence factor diagram (Figure 2C) suggested that the influence factors of microwave digestion of amino acids from large to small were the ratio of solid to liquid (C), microwave digestion temperature (A), and microwave digestion time (B). The 3D response surface diagram and 2D contour map of the interaction between the influencing factors could explain the interaction between the two factors. The slope was steeper and the color change was more obvious in 3D response surface plots, and the contour was flatter in 2D contour plots, indicating that the interaction between the two influencing factors was more significant [31,34]. The color and slope change of microwave digestion temperature (A) and microwave digestion time (B) was relatively moderate in the 3D response surface plot, and the contour line was closer to the ring in 2D contour plots, indicating that the interaction between them had little effect on the determination of amino acids (*p* = 0.8726, >0.05), as displayed in Figure 2D,G. The color and slope change of microwave digestion temperature (A) and solid–liquid ratio (C), microwave digestion time (B), and solid–liquid ratio (C) were steeper in the 3D response surface plot, and the contour line was closer to oblateness in the 2D contour plots, indicating that the interaction between them had a greater effect on the determination of amino acids (*p* = 0.0003 and *p* = 0.0006, <0.05), as presented in Figure 2E,H,F,I.

#### 3.2.4. Determination and Verification of the Conditions

Through the results of 17 groups of experiments at three factors and three levels in the BBD of RSM, the quadratic polynomial function of total amino acid yield, and microwave digestion temperature, microwave digestion time, solid–liquid ratio was used for nonlinear fitting and analysis of variance (ANOVA). The quadratic equation was established to predict the relationship between the digestion conditions and the yield of total amino acid. The optimum conditions were obtained as follows: microwave digestion temperature was 166.607 °C, microwave digestion time was 24.299 min, and the solid–liquid ratio was 46.591 g/mL. Under these conditions, the yield of total amino acids was 84.991 mg/g. According to the actual operation, the optimum conditions were slightly adjusted. The final conditions were 167 °C of the microwave digestion temperature, 24 min of the microwave digestion time, and 46.5 g/mL of the solid–liquid ratio. The actual yield of total amino acids was 84.296 mg/g, and the relative deviation between the actual and theoretical yield of total amino acids was 0.82%, indicating that the obtained fitting equation could accurately describe the relationship between the microwave digestion conditions (microwave digestion temperature, microwave digestion time and solid–liquid ratio) and the yield of total amino acid. Therefore, the optimized microwave digestion conditions were predictive, feasible, and scientific to measure QF’s total amino acid content through the BBD of RSM.

### 3.3. Method Validation

#### 3.3.1. System Suitability Test

According to the chromatographic conditions under “2.4”, the 17 amino acids in the standard solution and test solution were analyzed individually, as displayed in Figure 1a,e. The resolution between adjacent chromatographic peaks of 17 amino acids ranged from 1.261 to 7.178. It reflected that all chromatographic peaks could be well separated. In addition, the numbers of theoretical plates ranged from 3433.7 to 68,743.1, and the chromatographic peaks of each amino acid were relatively narrow, indicating that the separation effect of the chromatographic column was better.

#### 3.3.2. Linearity, LOD, and LOQ

The amount of 2.5 μmol/mL mixed amino acid stock solution was accurately measured and diluted with 0.01 mol/L hydrochloric acid to obtain 5, 10, 20, 40, 80, and 150 nmol/mL amino acid series standard solutions. Under the chromatographic conditions of “2.4”, the standard curve of the regression equation was fitted with the mass concentration *x* (μg/mL) as the abscissa and the peak area *y* as the ordinate. When the signal-to-noise ratio (S/N) of each amino acid content was 3, as the limit of detection (LOD), and the signal-to-noise ratio (S/N) was 10 as the limit of quantitative (LOQ). The mass concentration was converted by the molar mass of each amino acid. The results are presented in Table 5.

#### 3.3.3. Precision, Stability, and Repeatability

The mixed standard solution of 100 nmol/mL, including 17 amino acids, was measured 6 times successively. The RSD value of 17 amino acids ranged from 0.32% to 1.91%, indicating that the precision of the instrument was good. The sample S1 was accurately extracted and placed at room temperature for 0, 3, 6, 12, 18 and 24 h, respectively. The RSD value of 17 amino acids ranged from 0.45% to 4.01%, suggesting that the sample solution was stable. Sample S1 was prepared in parallel with 6 test solutions according to the “2.3” digestion conditions and measured according to the “2.4” chromatographic conditions. The RSD value of 17 amino acids ranged from 1.05% to 3.97%, confirming that the digestion method was repeatable, as listed in Table 6.

#### 3.3.4. Recovery

Six parallel samples (S1) of 0.1 g were weighed precisely into the digestion tube. An appropriate amount of mixed amino acid standard solution was added subsequently to ensure that the added concentration was 50 nmol/mL. Each test solution was digested according to the “2.3” digestion conditions and measured according to the “2.4” chromatographic conditions. The average recovery of 17 amino acids was 93.90~118.6%, and the RSD value was 0.46~3.82%, demonstrating that the accuracy of the method was reliable, as listed in Table 6.

### 3.4. Amino Acid Content and Composition

Based on the results of response surface optimization, 16 batches of QF were digested under the optimal digestion conditions of “2.3” and measured under the chromatographic conditions of “2.4”. Except for Try, which was not detected in SC2, SC3, and GX1, 17 amino acids were detected in all the batches. The content of total amino acids (TAA) in 16 batches was 71.88–91.03 mg/g, and the average content of total amino acids was 81.46 mg/g. The content of Glu (14.93–17.83 mg/g) and Arg (6.00–15.45 mg/g) were higher, and the content of Met (0–0.76 mg/g) and Cys (0.37–0.94 mg/g) was lower. The range of essential amino acids (EAA) content was 19.24–24.10 mg/g, and the range of non-essential amino acids (NEAA) content was 50.93–67.69 mg/g. Besides, the content range of medicinal amino acids (MAA), umami amino acids (UAA), sweet amino acids (SAA), and bitter amino acids (BAA) was 50.91–66.01 mg/g, 22.70~27.22 mg/g, 19.13–22.85 mg/g and 24.05–34.75 mg/g, respectively, as presented in Figure 3.

### 3.5. Nutritional Value Evaluation

When the composition and proportion of essential amino acids (EAA) are closer to the metabolic needs of the human body, they have higher nutritional value and bioavailability [35]. Through the above research, the ratio of essential amino acids to total amino acids (E/T) in QF ranged from 24.9% to 31.12%, with an average of 26.91%. The ratio of essential amino acids to non-essential amino acids (E/NE) in QF ranged from 33.24% to 45.18%, with an average of 36.88%, which was significantly lower than the ideal protein pattern (E/T = 40, E/NE = 60%) [13]. In addition, based on the standard pattern spectrum of essential protein amino acids referred to by the World Health Organization and Food and Agriculture Organization of the United Nations (FAO/WHO), as well as the ovalbumin pattern proposed by the Institute of Nutrition and Food Hygiene, Chinese Academy of Preventive Medicine, the essential amino acid composition and standard value of pattern spectrum in QF from different planted origins were compared [36]. The results showed that the content of essential amino acids in QF from different planted origins was lower than the standard values of the ovalbumin model. Only 4 batches of Val, 1 batch of Iso, 10 batches of Leu, and 4 batches of Phe + Tyr reached the FAO/WHO standard value among 16 batches, as presented in Table 7. Medicinal amino acids (MAA) were not fully synthesized by the human body, but they were necessary to maintain the body’s nitrogen balance [12]. The results indicated that the ratio of medicinal amino acids to total amino acids (M/T) in QF ranged from 67.80% to 73.20%, indicating that QF was rich in medicinal amino acids [13,37]. Among them, the contents of Glu and Arg were higher, accounting for 28.14% and 19.31% of the total medicinal amino acids, respectively. Glu is an important central neurotransmitter, which could lead to a variety of neurological diseases, such as Alzheimer’s disease, depression, epilepsy, and hepatic encephalopathy, when it is metabolically imbalanced [38,39]. Arg is a key amino acid involved in a variety of cellular processes and closely related to the regulation of body immunity, kidney disease, and cardiovascular disease [40,41]. Because QF contains neurotoxic active ingredients such as quisqualic acid and trigonelline [42], and the content of bitter amino acids in the QF was high. It reflected that its medicinal value was higher than its edible value.

### 3.6. Fingerprint Analysis

In order to compare the consistency of the composition of each amino acid between different planted origins, the fingerprint of similarity evaluation analysis was performed by importing the chromatography of 16 batches of QF at 570 nm and 440 nm into the Chinese Medicine Fingerprint Similarity Evaluation Software (2012 version). A total of 27 common peaks were matched, and 17 of them were identified by reference standards, which were Peak 1 (Asp), Peak 2 (Thr), Peak 3 (Ser), Peak 4 (Glu), Peak 6 (Gly), Peak 7 (Ala), Peak 8 (Cys), Peak 9 (Val), Peak 10 (Ile), Peak 11 (Leu), Peak 12 (Tyr), Peak 13 (Phe), Peak 16 (Lys), Peak 17 (NH3), Peak 18 (His), Peak 19 (Arg), and Peak 27 (Pro). The RSD ranges of retention time and peak area of each matched peak were 0.03–1.82% and 5.95–29.31%, respectively, and the range of similarity between the 16 batches of QF ranged from 0.889 to 0.999. The above analysis suggested that different planted origins of QF have a high degree of similarity. Therefore, it could provide a basis for the identification of QF species and quality consistency assessment according to the common amino acid ingredients, as presented in Figure 4a,b.

### 3.7. Chemometric Analysis

To compare the differences in amino acid content in QF from different planted origins and screen the important differential amino acids, the content of 17 amino acids in 16 batches of QF were introduced into SIMCA 14.1 for chemometric analysis [43]. Firstly, based on maintaining the original data, principal component analysis (PCA) was carried out to reduce dimensions. The model parameters R^2^ and Q^2^ were 0.844 and 0.554, respectively, indicating that the model interpretation rate was 88.4% and the prediction ability was 55.4%. The results suggested that QFs from different planted origins had a potential clustering tendency. (Figure 5a). Orthogonal partial least squares discriminant analysis (OPLS-DA) could further expand the differences between groups. The model parameters R^2^X, R^2^Y, and Q^2^ were 0.814, 0.618, and 0.538, respectively. The results demonstrated that QFs from different planted origins had an obvious clustering tendency (Figure 5b). It also reflected that QF has a similar geographical origin and content of various amino acids. R^2^ (0.0, 0.111) > 0 and Q^2^ (0.0, −0.362) < 0 for 200 permutation tests reflected that the model was reliable (Figure 5c). Meanwhile, cluster analysis (CA) was carried out to display clustering more intuitively, and it was consistent with the results of OPLS-DA. QF from different planted origins could be roughly divided into four categories according to the different amino acid content, which was consistent with their source of planted origins. The CQ and SC clustering were closer, while GX and YN clustering were closer (Figure 5d). In addition, six important differential amino acids were screened by variable importance projection (VIP > 1), which were Arg, Glu, Val, Asp, Ile, and Ser (Figure 5e). It reflected that the six amino acids played a vital role in the differences between different planted origins.

## 4. Conclusions

In this study, a pretreatment method of microwave-assisted digestion to determine amino acids in natural products of QF was established by response surface methodology. Compared with the traditional method, it has the advantages of shorter digestion time and the same reliable results. Meanwhile, the BBD experiments of RSM proved that the total amount of amino acids was the highest when the microwave digestion temperature was 167 °C, the microwave digestion time was 24 min, and the solid–liquid ratio was 46.5 g/mL. A total of 17 kinds of amino acid content in the natural products of QF from different planted origins were measured through optimal microwave-assisted digestion conditions combined with amino acid analyzer. Through the methodology validation, this method was fast, stable, and reliable, which was suitable to determine the amino acid profile in natural products of QF. The total amino acid content of QF from different planted origins ranged from 71.88 to 91.03 mg/g, and the amino acid composition and nutritional evaluation analysis indicated that it has a high medicinal nutritional value. In addition, fingerprint analysis showed that the similarity of different planted origins ranged from 0.889 to 0.999, while chemometrics analysis indicated that the QF from different planted origins had a various degree of clustering trend. In summary, our study establishes a rapid and convenient microwave digestion pretreatment method using a response surface methodology and measures and compares the amino acid profile of QF from different planted origins. It provided a reference for QF in the amino acid profile research, nutritional value evaluation, and geographical differentiation.

## Figures and Tables

**Figure 1 foods-13-01645-f001:**
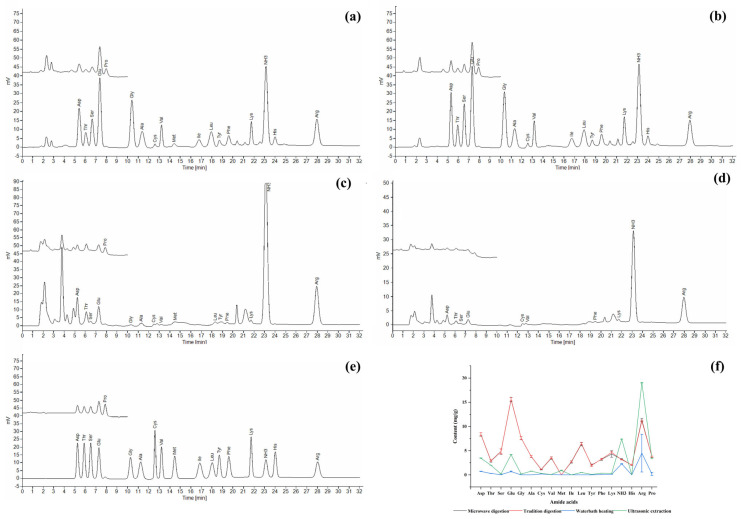
Chromatograms and amino acid contents comparison of various extraction methods. (**a**) Microwave digestion. (**b**) Tradition digestion. (**c**) Ultrasonic extraction. (**d**) Water bath heating. (**e**) Mixed standard solution of 17 amino acids. (**f**) Amino acid contents comparison of four extraction methods.

**Figure 2 foods-13-01645-f002:**
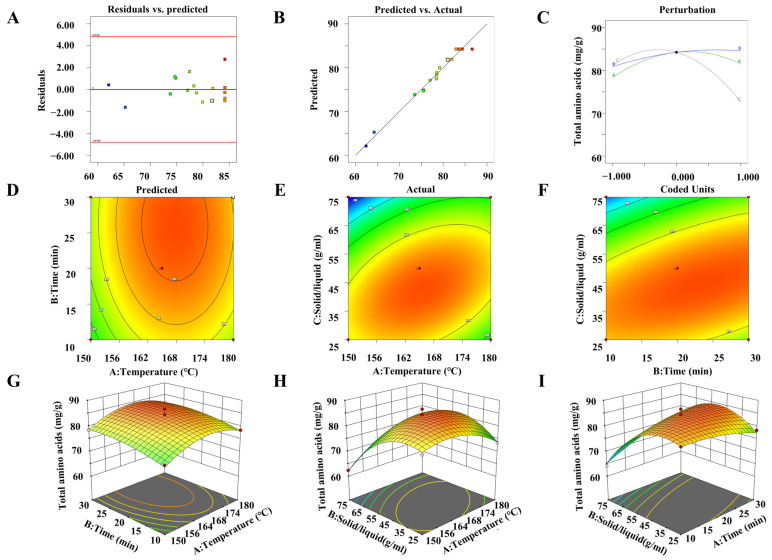
Box–Behnken model with tested variables. (**A**) Residuals vs. Predicted. (**B**) Predicted vs. actual. (**C**) Perturbation for RT determination. (**D**–**F**) 2D response surface. (**G**–**I**) 3D response surface.

**Figure 3 foods-13-01645-f003:**
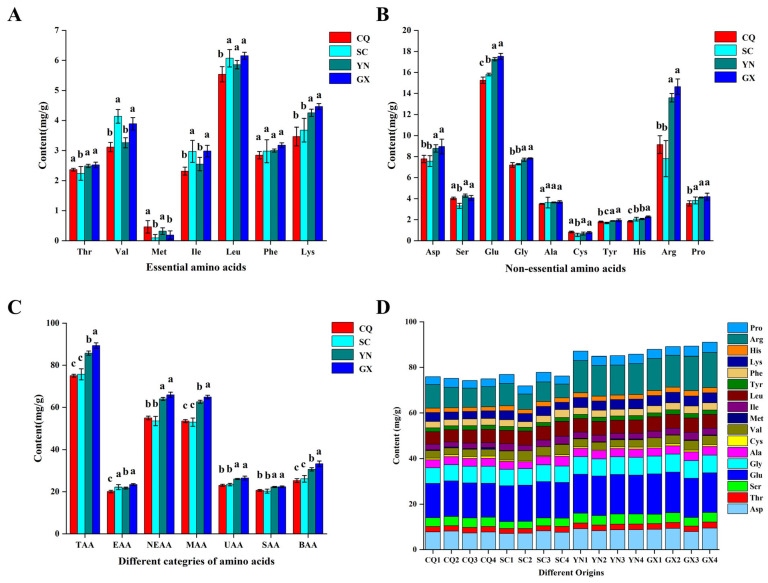
The analysis of amino acid content. (**A**) Essential amino acids. (**B**) Non-essential amino acids. (**C**) Different categories of amino acids. (**D**) The contents of total amino acids from various planted origins. The different character of a, b, c means significant (*p* < 0.05), and the same character of a, b, c means no significant (*p* > 0.5).

**Figure 4 foods-13-01645-f004:**
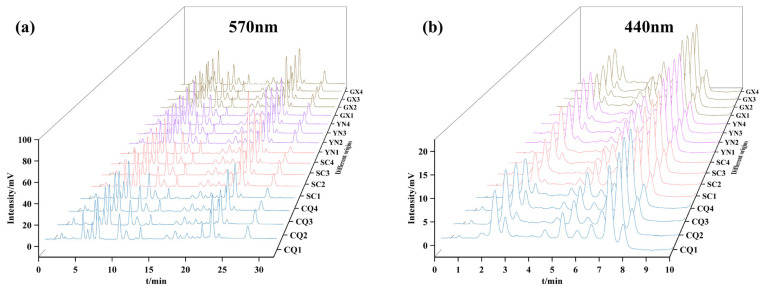
The fingerprint of amino acid in QF: (**a**) 570 nm; (**b**) 440 nm.

**Figure 5 foods-13-01645-f005:**
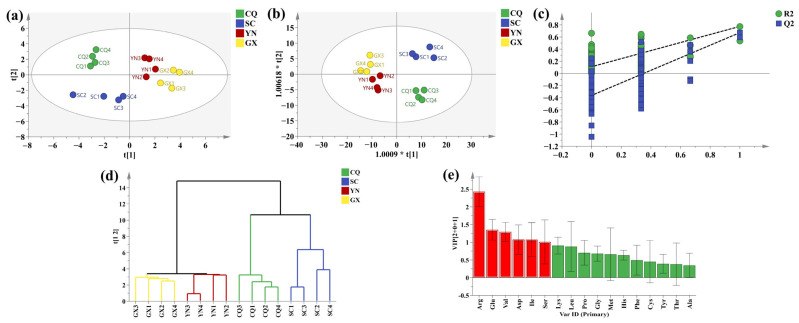
Chemometric analysis. (**a**) PCA score plot. (**b**) OPLS-DA score plot. (**c**) Permutation scatter plot. (**d**) HCA plot. (**e**) VIP values.

**Table 1 foods-13-01645-t001:** Microwave digestion conditions.

Step	Temperature/°C	Heating Time/min	Pressure/atm
1	80	2	10
2	100	2	20
3	120	2	30
4	140	2	40
5	167	24	40

**Table 2 foods-13-01645-t002:** Three-factor and three-level experimental design.

Level	A: Temperature/°C	B: Heating Time/min	C: Solid–Liquid Ratio/(g: mL)
−1	150	10	25
0	165	20	50
1	180	30	75

**Table 3 foods-13-01645-t003:** Box–Behnken design and results.

No.	A/°C	B/min	C/(g: mL)	Total Amino Acids (mg/g)
1	150	10	50	75.56
2	180	10	50	78.54
3	150	30	50	78.56
4	180	30	50	81.08
5	150	20	25	79.20
6	180	20	25	73.52
7	150	20	75	62.40
8	180	20	75	75.46
9	165	10	25	81.98
10	165	30	25	78.47
11	165	10	75	64.24
12	165	30	75	77.05
13	165	20	50	86.65
14	165	20	50	83.84
15	165	20	50	83.20
16	165	20	50	84.44
17	165	20	50	82.95

**Table 4 foods-13-01645-t004:** Results of variance analysis.

Source	Sum of Squares	Degrees of Freedom	Mean Square	F-Value	*p*-Value	Significance
Model	682.07	9	75.79	39.97	<0.0001	**
A	20.70	1	20.70	10.92	0.0130	**
B	27.54	1	27.54	14.52	0.0066	**
C	144.59	1	144.59	76.25	<0.0001	**
AB	0.0524	1	0.0524	0.0277	0.8726	-
AC	87.83	1	87.83	46.32	0.0003	**
BC	66.54	1	66.54	35.09	0.0006	**
A^2^	77.40	1	77.40	40.82	0.0004	**
B^2^	9.39	1	9.39	4.95	0.0614	*
C^2^	223.38	1	223.38	117.80	<0.0001	**
Residual	13.27	7	1.90			
Lack of fit	4.54	3	1.51	0.6928	0.6027	-
Pure error	8.73	4	2.18			
Total	695.35	16				

Note: “-” means no significance; “*” means significance (*p* < 0.05); “**” means extremely significant (*p* < 0.01).

**Table 5 foods-13-01645-t005:** The linearity, correlation coefficient, linear range, LODs, and LOQs of 17 amino acids.

Rt (min)	Amino Acids	Linear Range (nmol/mL)	Linear Equation	*r*	LODs (μg/mL)	LOQs (μg/mL)
5.23	Asp	5–150	*y* = 23.12*x* − 1.827	0.9999	0.015	0.049
5.88	The	5–150	*y* = 25.75*x* − 1.107	0.9999	0.013	0.043
6.49	Ser	5–150	*y* = 29.88*x* − 1.824	0.9999	0.011	0.038
7.25	Glu	5–150	*y* = 21.19*x* − 0.8424	0.9999	0.018	0.061
7.87	Pro	5–150	*y* = 9.417*x* − 2.118	0.9999	0.014	0.047
10.33	Gly	5–150	*y* = 35.30*x* − 1.767	0.9999	0.019	0.063
11.29	Ala	5–150	*y* = 29.23*x* + 1.820	0.9998	0.018	0.061
12.59	Cys	5–150	*y* = 13.13*x* + 1.784	0.9998	0.015	0.050
13.19	Val	5–150	*y* = 23.19*x* − 1.771	0.9999	0.027	0.090
14.38	Met	5–150	*y* = 18.09*x* − 2.271	0.9998	0.039	0.131
16.75	Ile	5–150	*y* = 20.73*x* − 4.021	0.9999	0.029	0.096
17.93	Leu	5–150	*y* = 16.37*x* + 4.008	0.9996	0.026	0.087
18.67	Tyr	5–150	*y* = 12.37*x* + 2.436	0.9994	0.034	0.113
19.57	Phe	5–150	*y* = 16.69*x* − 5.923	0.9999	0.013	0.045
21.73	Lys	5–150	*y* = 22.81*x* − 2.337	0.9998	0.022	0.074
23.97	His	5–150	*y* = 17.91*x* − 1.476	0.9999	0.050	0.165
27.97	Arg	5–150	*y* = 15.21*x* − 3.567	0.9999	0.154	0.512

**Table 6 foods-13-01645-t006:** The precision, stability, repeatability, and recovery of 17 amino acids.

Amino Acids	Precision (RSD%)	Stability (RSD%)	Repeatability (RSD%)	Recovery	
				Average Recovery/%	RSD%
Asp	0.47	0.81	1.05	112.9	2.15
Thr	0.34	4.01	1.38	107.4	1.46
Ser	0.32	0.84	1.45	109.9	3.82
Glu	0.34	0.50	1.22	107.5	1.42
Pro	0.78	0.58	1.39	115.2	0.93
Gly	1.07	0.68	1.86	104.4	0.51
Ala	0.70	2.45	3.04	97.07	1.41
Cys	0.73	0.76	2.11	102.8	3.25
Val	0.68	1.81	2.44	97.07	1.65
Met	0.56	0.86	2.99	102.6	2.83
Ile	1.91	0.84	1.55	99.33	0.94
Leu	1.70	3.10	2.64	103.3	1.66
Tyr	0.58	0.45	3.97	102.4	0.50
Phe	0.45	0.86	3.95	104.1	0.46
Lys	0.54	0.79	1.45	101.7	0.57
His	0.40	0.91	1.20	110.2	1.30
Arg	0.36	1.25	1.55	109.3	0.88

**Table 7 foods-13-01645-t007:** Nutritional value evaluation of QF.

No.	Thr	Val	Met+Cys	Ile	Leu	Phe+Tyr	Lys	Total
CQ1	3.10	4.33	1.43	3.26	7.14	5.91	5.18	30.36
CQ2	3.07	4.00	1.69	2.99	6.97	5.94	4.40	29.07
CQ3	3.25	4.31	1.51	3.19	7.73	6.50	4.43	30.93
CQ4	3.18	3.95	2.28	2.90	7.69	6.38	4.45	30.84
SC1	2.75	5.16	0.95	3.99	7.50	6.04	5.15	31.53
SC2	2.87	5.89	0.74	3.45	8.55	5.76	4.85	32.11
SC3	2.85	5.06	0.47	3.84	7.63	5.97	5.21	31.03
SC4	3.38	5.79	1.28	4.41	8.45	6.85	4.23	34.39
YN1	2.93	4.02	1.09	3.05	6.77	5.61	5.01	28.49
YN2	2.96	3.72	0.79	3.31	7.11	5.80	4.81	28.51
YN3	2.88	3.68	1.36	2.73	6.71	5.64	5.01	28.02
YN4	2.84	3.77	1.42	2.81	6.74	5.73	5.03	28.33
GX1	2.86	4.32	0.78	3.27	7.10	5.56	5.11	29.00
GX2	2.84	4.17	1.15	3.19	6.82	5.78	5.16	29.11
GX3	2.70	4.69	1.18	3.65	7.02	5.92	4.92	30.07
GX4	2.90	4.24	1.27	3.24	6.63	5.65	4.81	28.74
FAO/WHO	4.0	5.0	3.5	4.0	7.0	6.0	5.5	35
Whole Egg	5.1	8.8	6.4	6.6	10.0	7.3	5.5	49.7

## Data Availability

The original contributions presented in the study are included in the article, further inquiries can be directed to the corresponding author.

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
