# Peer review of "An Optimized Microwave-Assisted Digestion Method to Analyze the Amino Acids Profile of *Quisqualis Fructus* from Different Planted Origins"

_foods, 2024, doi:10.3390/foods13111645_

Round 1

Reviewer 1 Report

Comments and Suggestions for Authors

Small findings are reported directly in the attached text

Comments on the Quality of English Language

Author Response

Reviewer#1

Comments 1: L85: The discussion on the present literature and finding out the research gap is missing here. It is required to establish the novelty of this work.

Response 1: We agree with this comment. Therefore, we have discussed the novelty of our work. As displayed in L81-84 and L86-L88.

Comments 2: L125: Please check.

Response 2: The space has been deleted.

Comments 3: L118 and L134: The solid-to-solvent ratio differs in these cases. Please justify.

Response 3: We explored the amino acid extraction method before response surface optimization. Therefore, the material-liquid ratios of these four methods that 0.2 g of sample was added to 10 ml of digestion solution were consistent in making methodological comparisons. We have justified for a clearer presentation. As displayed in “2.3” section.

Comments 4: L157-158: Please justify the upper and lower limits of these conditions.

Response 4: According to the previous report by the same manufacturer of the digestion apparatus, the microwave conditions were used for milk powder digestion [1]. Therefore, we justified the upper and lower limits of these conditions refer to the relevant literature reports and pre-experimental results [2,3]. Besides, the optimal microwave-assisted digestion conditions in the range of upper and lower limits of these conditions were found subsequently through response surface optimization.

[1]     Lu, Y.L.; Guo, D.H.; Ni, C.J.; Jin, X.J.; Deng, X.J. A method for rapid hydrolysis of protein in dairy products by an inflatable microwave digestion device. CN112985970B, Shanghai, 2021.

[2]     Weber, P. Determination of amino acids in food and feed by microwave hydrolysis and UHPLC- MS/MS. J Chromatogr B Analyt Technol Biomed Life Sci. 2022, 1209, 123429.

[3]     He, Q.; Lei, Q.; Huang, S.; Zhou, Y.; Liu, Y.; Zhou, S.; Peng, D.; Deng, X.; Xue, J.; Li, X.; Qiu, H. Effective extraction of bioactive alkaloids from the roots of Stephania tetrandra by deep eutectic solvents-based ultrasound-assisted extraction. J Chromatogr A. 2023, 1689, 463746.

Comments 5: L163-164: Please elaborate more on the optimization technique used.

Response 5: We have elaborated more on the optimization technique used. As displayed in “2.6” section.

Comments 6: L179-182: Please check the sentence formation.

Response 6: We have modified the sentence formation. As displayed in L193-194.

Comments 7: L198-199: Not relevant.

Response 7: We modified the inappropriate expression. As displayed in L210-211.

Comments 8: Table 3: Please perform a significance test on the response – total amino acids (indicator) data set across the column. Is there any requirement for significant digits up to 4 decimal places?

Response 8: According to your suggestion, we found that there is no significant difference in the results after retaining four significant digits to two decimal places. Therefore, we kept the four significant digits up to two decimal places based on the requirement of less than 0.1% analytical error. As displayed in Table 3.

Comments 9: L199-200: Please note that RSM is not an optimization method. The methodology of RSM ends once you have developed the model and generated the response surface plot. You have used numerical optimization, an additional tool with experimental design, and RSM.

Response 9: We have revised the expression of BBD model of RSM. As displayed in L211-214.

Comments 10: L204: Software can't be applied; you have to instruct it what to do. So, please discuss the decision-making steps you employed here.

Response 10: We have discussed the decision-making steps. As displayed in L218-221.

Comments 11: L205: Why was only a quadratic polynomial model developed? Whyn't other models be tested? How the best-fitted model was selected?

Response 11:  We adopt the common quadratic polynomial model and obtained satisfactory results with reliable predictive ability and high goodness of fit [1,2]. Therefore, the other models were not adopted in our research.

[1]     Khruengsai S, Promhom N, Sripahco T, Siriwat P, Pripdeevech P. Optimization of enzyme-assisted microwave extraction of Zanthoxylum limonella essential oil using response surface methodology. Sci Rep. 2023 Aug 8;13(1):12872.

[2]     Zhang M, Wei D, He L, Wang D, Wang L, Tang D, Zhao R, Ye X, Wu C, Peng W. Application of response surface methodology (RSM) for optimization of the supercritical CO2 extract of oil from Zanthoxylum bungeanum pericarp: Yield, composition and gastric protective effect. Food Chem X. 2022 Jul 11;15: 100391.

Comments 12: L208: Is the equation in the coded form of factors or the real form of factors?

Response 12: The equation was in the coded form of factors, it has been modified in the resubmitted manuscript. As displayed in L221.

Comments 13: L211: What about the lack of fit test for the model?

Response 13: The information of lack of fit test has been added in the resubmitted manuscript. As displayed in L223-L224.

Comments 14: L222-239: Discussion with square terms of A and C is not present here. Please elaborate on it with respect to the shape of the contours obtained in that landscape.

Response 14: We accepted this suggestion and discussed the square terms of A and C. Meanwhile, we discussed more detailed in this part. As displayed in L243-L253.

Comments 15: Figure 2: Only contours are enough to represent the trend.

Response 15: We explained more detail about 2D and 3D plots to represent their trend. Meanwhile, we think that both 2D and 3D plots could represent the trend refer to relevant literature. As displayed in L243-L253.

[1]     Sk, M.H.; Habibur, R.; Nafisur, R.; Syed, N.H.A.; Omar, A.; Saikh, M.W.; Masoom, R.S.; Mahboob, A. Application of Box–Behnken design combined response surface methodology to optimize HPLC and spectrophotometric techniques for quantifying febuxostat in pharmaceutical formulations and spiked wastewater samples. Microchemical Journal. 2023, 184, 108191.

[2]     He, Q.; Lei, Q.; Huang, S.; Zhou, Y.; Liu, Y.; Zhou, S.; Peng, D.; Deng, X.; Xue, J.; Li, X.; Qiu, H. Effective extraction of bioactive alkaloids from the roots of Stephania tetrandra by deep eutectic solvents-based ultrasound-assisted extraction. J Chromatogr A. 2023, 1689, 463746.

Comments 16: L244: Please elaborate more on the optimization process. Which type of optimization was used in this case? Don’t say that it has been optimized using so-and-so software. Please remember that software is just a tool, and you have instructed it to operate for optimization. Therefore, the fundamentals should be clear in this aspect and presented properly in the manuscript. Please report the type of objective function.

Response 16: We elaborated more on the optimization process, including the fundamentals and type of objective function. As displayed in L260-L262.

Comments 17: L247: What do you mean by comprehensive score? Is it the desirability value?

Response 17: Thank you for your careful review. The inaccurate expression has been revised. As displayed in L264-L269.

Comments 18: L267-274: It seems more like a methodology part.

Response 18: Thank you for pointing this out. It's really part of the methodology including Linearity, LOD and LOQ.

Comments 19: Table 5: What do you think about linear equations in the fourth column? How is the presence of an intercept justified? Say: peak area (y) = 23.12 x concentration+1.827. Now, if you put x=0, it means that even though there is no solute, there will be a peak area of 1.827 units. Please look into it.

Response 19: Each amino acids were detected at visible wavelengths (570nm and 440nm) after a colorimetric reaction with ninhydrin reagent in the post-column reaction chamber. Therefore, it’s inevitable to result in a higher matrix background and lead to a larger intercept.

Large intercepts in linear equations for amino acid analysis are a very common phenomenon [1,2]. At present, the external standard one-point method was commonly performed for quantification in the amino acid determination by amino acids analyzer. Therefore, standard curve will not affect the final test results whether the linearity of the method is good or not.

In the case of the sample concentration and the control concentration is relatively close,

In the case of the sample concentration and the control concentration is relatively close, the external standard one-point method of quantification could meet the requirements even if the intercept of the standard curve is larger. Besides, the results of recovery test showed that the method of quantification meet the requirement.

[1]     Ping, Y.W.; Fei, F.S.; Jia, X.Y; Yu, X.J.; Run, Z.H.; Tao, C.; Xiao, H.Y.; Wei, G.Z; Li, L.; Dong, Y.Z. A rapid and efficient method of microwave-assisted extraction and hydrolysis and automatic amino acid analyzer determination of 17 amino acids from mulberry leaves. Industrial Crops & Products. 2022, 186,115271.

[2]     Bhandari, S.D.; Gallegos-Peretz, T.; Wheat, T.; Jaudzems, G.; Kouznetsova, N.; Petrova, K.; Shah, D.; Hengst, D.; Vacha, E.; Lu, W.; Moore, J.C.; Metra, P.; Xie, Z. Amino Acid Fingerprinting of Authentic Nonfat Dry Milk and Skim Milk Powder and Effects of Spiking with Selected Potential Adulterants. Foods. 2022, 11,2868.

Comments 20: L292: What is the benchmark for reliable and accurate methods?

Response 20: The benchmark was that all the results of methodological validation were in accordance with the guidelines for analytical method validation of the Chinese Pharmacopoeia (2020 edition).

Comments 21: Please add the significance test in Figure 3.

Response 21: We have added the significance test in Figure 3. The different character of a, b, c means significant (p<0.05) and the same character of a, b, c means no significant (p>0.5).

Comments 22: L332: Please add some literature support for the trend obtained.

Response 22: Thank you for pointing this out. We have added relevant literature in resubmitted manuscript.  As displayed in L352.

Comments 23: L352: Is it possible to present the %area matched for each?

Response 23: Thank you for pointing this out. The Chinese Medicine Fingerprint Similarity Evaluation Software (2012 version) was able to display the %area of each matched peak.

Comments 24: Section 3.6: Does the author think employing multiway analysis might be a better option here? It's just a suggestion.

Response 24: In order to control the quality of Chinese medicines in a more comprehensive way, the fingerprints of Chinese medicines was established to identify the common ingredients and to reflect the totality of the Chinese medicines. Meanwhile, it ensured the stability and reliability of the Chinese medicinal preparations from the perspective of the chemical substance base [1,2]. Therefore, it could provide a basis for the identification of QF species and quality consistency assessment according to the common amino acid ingredients.

[1]     Xiao Y, Shan X, Wang H, Hong B, Ge Z, Ma J, Li Y, Zhao Y, Ma G, Zhang C. Spectrum-effect relationship between HPLC fingerprint and antioxidant of "San-Bai Decoction" extracts. J Chromatogr B Analyt Technol Biomed Life Sci. 2022 Oct 1;1208: 123380.

[2]     Wang D, Gu X, Fang K, Fu B, Liu Y, Di X. Study on quality control of Zuojin pill by HPLC fingerprint with quantitative analysis of multi-components by single marker method and antioxidant activity analysis. J Pharm Biomed Anal. 2023 Feb 20;225: 115075.

Comments 25: L355: Please elaborate more on your recommendations.

Response 25: We have supplied some recommendations in resubmitted manuscript. As displayed in L374-L375.

Comments 26: L365: Prediction ability was low.

Response 26: There is no doubt that the prediction ability of 55.4% was low, but it’s meeting the requirements of the model when this value was greater than 0.5 [1,2].

[1]     Qie M, Li Y, Hu X, Zhaxi C, Zhao S, Zhang Z, Yang X, Bai L, Zhao Y. A New and Effective Method to Trace Tibetan Chicken by Amino Acid Profiling. Foods. 2023 Feb 18;12(4):876.

[2]     Wang M, Lee J, Zhao J, Chatterjee S, Chittiboyina AG, Ali Z, Khan IA. Comprehensive quality assessment of peppermint oils and commercial products: An integrated approach involving conventional and chiral GC/MS coupled with chemometrics. J Chromatogr B Analyt Technol Biomed Life Sci. 2024 Jan 1;1232: 123953.

Comments 27: Please discuss more from the clusters obtained in Figure 5b.

Response 27: We have discussed more about the clusters obtained in Figure 5b. As displayed in L391-L392.

Comments 28: L383-403: Please make it short and add some quantifiable conditions. Please refrain from extrapolation.

Response 28: We have simplified the conclusion part.

Reviewer 2 Report

Comments and Suggestions for Authors

L85: The discussion on the present literature and finding out the research gap is missing here. It is required to establish the novelty of this work.

L125: Please check.

L118 and L134: The solid-to-solvent ratio differs in these cases. Please justify.

L157-158: Please justify the upper and lower limits of these conditions.

L163-164: Please elaborate more on the optimization technique used.

L179-182: Please check the sentence formation.

L198-199: Not relevant.

Table 3: Please perform a significance test on the response – total amino acids (indicator) data set across the column. Is there any requirement for significant digits up to 4 decimal places?

L199-200: Please note that RSM is not an optimization method. The methodology of RSM ends once you have developed the model and generated the response surface plot. You have used numerical optimization, an additional tool with experimental design, and RSM.

L204: Software can't be applied; you have to instruct it what to do. So, please discuss the decision-making steps you employed here.

L205: Why was only a quadratic polynomial model developed? Whyn't other models be tested? How the best-fitted model was selected?

L208: Is the equation in the coded form of factors or the real form of factors?

L211: What about the lack of fit test for the model?

L222-239: Discussion with square terms of A and C is not present here. Please elaborate on it with respect to the shape of the contours obtained in that landscape.

Figure 2: Only contours are enough to represent the trend.

L244: Please elaborate more on the optimization process. Which type of optimization was used in this case? Don’t say that it has been optimized using so-and-so software. Please remember that software is just a tool, and you have instructed it to operate for optimization. Therefore, the fundamentals should be clear in this aspect and presented properly in the manuscript. Please report the type of objective function.

L247: What do you mean by comprehensive score? Is it the desirability value?

L267-274: It seems more like a methodology part.

Table 5: What do you think about linear equations in the fourth column? How is the presence of an intercept justified? Say: peak area (y) = 23.12 x concentration+1.827. Now, if you put x=0, it means that even though there is no solute, there will be a peak area of 1.827 units. Please look into it.

L292: What is the benchmark for reliable and accurate methods?

Please add the significance test in Figure 3.

L332: Please add some literature support for the trend obtained.

L352: Is it possible to present the %area matched for each?

Section 3.6: Does the author think employing multiway analysis might be a better option here? It's just a suggestion.

L355: Please elaborate more on your recommendations.

L365: Prediction ability was low.

Please discuss more from the clusters obtained in Figure 5b.

L383-403: Please make it short and add some quantifiable conditions. Please refrain from extrapolation.

Author Response

Reviewer#2

Comments 1: It would be helpful to include a map of the sampling locations.

Response 1: We have added the sampling locations in China in the abstract plot.

Comments 2: Some numerical and sentence problems in text.

Response 2: We have checked and revised the corresponding part of full text. As displayed in resubmitted manuscript.

4. Response to Comments on the Quality of English Language

Point 1: None

Response 1: We have examined the English expression of the full text again.

5. Additional clarifications

Thank the reviewers again for their valuable opinions. The detail information was displayed in the resubmitted manuscripts with red-colored text.

Round 2

Reviewer 2 Report

Comments and Suggestions for Authors

The authors have revised the manuscript with respect to most of the suggestions and queries provided. However, for some comments, the answers were satisfactory, but the corresponding amendment has not been performed in the manuscript. It is also recommended that the replies be suitably included in the manuscript. The readers may also have the same doubts. Those comments are as follows.

-Please justify the upper and lower limits of these conditions. 

-Please elaborate more on the optimization technique used.

-Why was only a quadratic polynomial model developed? Whyn't other models be tested? How the best-fitted model was selected?

-Please elaborate more on the optimization process. Which type of optimization was used in this case? Please report the type of objective function.

Author Response

Response to Reviewer X Comments 1. Summary Thank you very much for taking the time to review this manuscript. Please find the detailed responses below and the corresponding revisions/corrections highlighted/in track changes in the re-submitted files, which has been marked with blue text. 2. Questions for General Evaluation Reviewer’s Evaluation Response and Revisions Does the introduction provide sufficient background and include all relevant references? Yes I agree with this decision. Are all the cited references relevant to the research? Yes I agree with this decision. Is the research design appropriate? Yes I agree with this decision. Are the methods adequately described? Can be improved I agree with this decision. Inadequate areas have been revised. Are the results clearly presented? Can be improved I agree with this decision. Are the conclusions supported by the results? Yes I agree with this decision. 3. Point-by-point response to Comments and Suggestions for Authors Reviewer#1 Comments 4: L157-158: Please justify the upper and lower limits of these conditions. Response 4: We elaborate more detail to justify the upper and lower limits. As displayed in L162-L167. Comments 5: L163-164: Please elaborate more on the optimization technique used. Response 5: We have elaborated more on the optimization technique used. As displayed in “2.6” section. Comments 11: L205: Why was only a quadratic polynomial model developed? Whyn't other models be tested? How the best-fitted model was selected? Response 11: In order to simplify the regression model and avoid overfitting, the quadratic polynomial model was developed. Meanwhile, we adopt the common quadratic polynomial model and obtained satisfactory results with reliable predictive ability and high goodness of fit [1,2]. Therefore, the other models were not adopted in our research. As displayed in L228-L231 and L243-L246. Comments 16: L244: Please elaborate more on the optimization process. Which type of optimization was used in this case? Don’t say that it has been optimized using so-and-so software. Please remember that software is just a tool, and you have instructed it to operate for optimization. Therefore, the fundamentals should be clear in this aspect and presented properly in the manuscript. Please report the type of objective function. Response 16: The objective function was quadratic polynomial function. We elaborated more on the optimization process about of getting the optimal result. As displayed in L276-L279. 4. Response to Comments on the Quality of English Language Point 1: None Response 1: We have examined the English expression of the full text again. 5. Additional clarifications Thank the reviewers again for their valuable opinions. The insufficient part was modified again with blue text. The detail information was displayed in the resubmitted manuscripts.
